# Accuracy of Ultrasonography and Magnetic Resonance Imaging for Preoperative Staging of Cervical Cancer—Analysis of Patients from the Prospective Study on Total Mesometrial Resection

**DOI:** 10.3390/diagnostics11101749

**Published:** 2021-09-23

**Authors:** Maciej Stukan, Paul Buderath, Bartosz Szulczyński, Jacek Gębicki, Rainer Kimmig

**Affiliations:** 1Department of Gynecologic Oncology, Gdynia Oncology Center, Pomeranian Hospitals, ul. Powstania Styczniowego 1, 81519 Gdynia, Poland; 2Division of Propedeutics of Oncology, Medical University of Gdańsk, ul. Powstania Styczniowego 9B, 81519 Gdynia, Poland; 3West German Cancer Center, Department for Gynecology and Obstetrics, University Hospital Essen, University of Duisburg-Essen, Hufelandstraße 55, 45147 Essen, Germany; paul.buderath@uk-essen.de (P.B.); Rainer.Kimmig@uk-essen.de (R.K.); 4Department of Process Engineering and Chemical Technology, Faculty of Chemistry, Gdańsk University of Technology, ul. Narutowicza 11/12, 80233 Gdańsk, Poland; bartosz.szulczynski@pg.edu.pl (B.S.); jacek.gebicki@pg.edu.pl (J.G.)

**Keywords:** cervical cancer, ultrasound, magnetic resonance imaging, accuracy, specificity, total mesometrial resection, staging, uterine corpus, parametrium

## Abstract

We aimed to evaluate the accuracy of ultrasonography with gynecologic examination performed by a gynecological oncologist and magnetic resonance imaging (MRI) interpreted by a radiologist for the local and regional staging of patients with early-stage cervical cancer. The study was a single-site sub-analysis of the multi-institutional prospective, observational Total Mesometrial Resection (TMMR) Register Study, which included all consecutive study patients from Gdynia Oncology Center. Imaging results were compared with pathology findings. A total of 58 consecutive patients were enrolled, and 50 underwent both ultrasonography and MRI. The accuracy of tumor detection and measurement errors was comparable across ultrasonography and MRI. There were no significant differences between ultrasonography and MRI in the accuracy of detecting parametrial involvement (92%, confidence interval (CI) 84–100% vs. 76%, CI 64–88%, *p* = 0.3), uterine corpus infiltration (94%, CI 87–100% vs. 86%, CI 76–96%, *p* = 0.3), and vaginal fornix involvement (96%, CI 91–100% vs. 76%, CI 64–88%, *p* = 0.3). The importance of uterine corpus involvement for the first-line lymph node metastases was presented in few cases. The accuracy of ultrasonography was higher than MRI for correctly predicting tumor stage: International Federation of Gynecology and Obstetrics (FIGO)–2018: 69%, CI 57–81% vs. 42%, CI 28–56%, *p* = 0.002, T (from TNM system): 79%, CI 69–90% vs. 52%, CI 38–66%, *p* = 0.0005, and ontogenetic tumor staging: 88%, CI 80–96% vs. 70%, CI 57–83%, *p* = 0.005. For patients with cervical cancer who are eligible for TMMR and therapeutic lymphadenectomy, the accuracy of ultrasonography performed by gynecological oncologists is not inferior to that of MRI interpreted by a radiologist for assessing specific local parameters, and is more accurate for local staging of the disease and is thus more clinically useful for planning adequate surgical treatment.

## 1. Introduction

According to recent guidelines for the management of patients with cervical cancer, the mandatory initial workup to assess pelvic tumor extent and guide treatment options should be conducted using pelvic magnetic resonance imaging (MRI). Endovaginal/transrectal ultrasound is an option if performed by a properly trained sonographer [1].

MRI data are typically analyzed and described by the radiologist, who does not see the patient. Furthermore, MRI is a time-consuming and costly procedure that requires planning. In most cases, the examination is performed using a contrast medium, and results are evaluated by the radiologist. However, experienced sonographers can obtain accuracies for local staging of patients with early cervical cancer using ultrasonography that are better than or comparable to those obtained by MRI [2,3,4,5,6]. Even in patients with more advanced disease who are referred for primary definitive chemoradiotherapy, ultrasonography can be as accurate as MRI [7,8,9]. Thus, ultrasonography can be considered an alternative imaging method to MRI for the local (and regional) staging of cervical cancer. Some institutions no longer perform MRI, whereas others use both ultrasonography and MRI before surgery. Ultrasonography is a standard element of gynecological examination and plays an important clinical role in the preoperative work-up of patients with cervical cancer. The examination can be performed by a gynecological oncologist, who also performs a clinical examination and administers further treatment. Ultrasound machines are easily available, and no special patient preparation is required before the exam. Therefore, if the accuracy of ultrasound is comparable to or even better than that of MRI, the patient may be spared from undergoing an MRI examination, which would also lower the cost of diagnostic imaging.

Treatment options for patients with newly diagnosed cervical cancer are surgery (radical hysterectomy and systematic pelvic and periaortic lymphadenectomy) or definitive radiochemotherapy [1]. Correctly assigning the treatment modality is crucial and is based on clinical examination and imaging. To minimize the risk of treatment-related complications, the combination of surgery and irradiation should be avoided. However, some patients who undergo surgery exhibit risk factors (e.g., positive lymph nodes (LNs), parametrial involvement, and significant cervical stromal invasion), which are indications for adjuvant radiotherapy.

Total mesometrial resection (TMMR) is a surgical procedure that was developed [10], implemented, and verified [11] by Hockel et al. for the treatment of patients with cervical cancer. TMMR excises the entire Müllerian compartment, except the distal vagina, and the vascular and ligamentous mesometria. Therapeutic lymph node dissection (tLND) is performed alongside TMMR, which involves harvesting the first-, second-, and third-line LN regions of cervical cancer. The type of surgery selected for each patient is guided by ontogenetic tumor (oT) and nodal stages, which are determined pre- and intraoperatively. Adjuvant radiotherapy is not required for patients who undergo TMMR/tLND. Although not a substitute for a randomized trial, oncological results of a prospective study on cancer field surgery without adjuvant radiotherapy were favorable to those of studies on traditional surgery and adjuvant chemoradiotherapy in terms of histopathological risk factors and primary chemoradiotherapy [12,13,14,15,16,17].

The primary objectives of the current study were to evaluate the accuracy of ultrasonography performed by a gynecological oncologist for the local staging of cervical cancer and to compare the accuracy of ultrasonography (performed by a gynecological oncologist) and MRI (interpreted by a radiologist) for the local staging of cervical cancer. The secondary optional objective was to evaluate the accuracy of ultrasonography and MRI for the regional staging of cervical cancer.

## 2. Materials and Methods

### 2.1. Study Design

This was a single-site sub-analysis of the prospective, observational TMMR Register Study (TMMR-RS; NCT01819077), which included all consecutive study patients from the Gdynia Oncology Center. The design of the TMMR-RS allowed the acquisition of high-quality clinical and histological data, which enabled the reliable analysis of preoperative imaging accuracy. Although not a requirement of the TMMR-RS, all patients received preoperative ultrasound examination as part of their routine workup, and results were collected at the investigation site. Imaging data were prospectively collected before surgery, saved, and remained unchanged thereafter. Data on the TMMR/tLND procedure and histology results were collected and saved after the procedure.

Study data were collected and managed using REDCap electronic data capture tools, hosted at Pomeranian Hospitals, Gdynia Oncology Center, Poland [18,19]. No patient-identifiable data were collected. A time interval of up to 30 days between the ultrasound and MRI examinations and surgery was permitted for patients to be included in the study.

### 2.2. Participants

Patients included in this study had histologically proven stages IB–IIA (preoperatively, according to the International Federation of Gynecology and Obstetrics [FIGO]; no restrictions on tumor size) cervical cancer (squamous cell carcinoma or adenocarcinoma), with a Karnofsky index of ≥70 and unrestricted operability. Patients had a body mass index (BMI) of <35 and were aged ≥ 18 years. Patient treatment decisions of TMMR and tLND without adjuvant radiotherapy were made by the responsible clinic (clinician) on a routine clinical basis. Informed consent for participation in the TMMR-RS was obtained from all patients (https://clinicaltrials.gov/ct2/show/NCT01819077, accessed on 29 July 2021).

TMMR excises the complete Müllerian compartment, except the distal vagina, and the vascular and ligamentous mesometria [11]. TMMR and tLND procedures were performed with strict adherence to the methodology described elsewhere [11,20,21]. The lymphadenectomy procedure was slightly modified. It began with periaortic inframesenterial tLND, which were sent for frozen sectioning; in the cases of negative results, lower levels of LNs were removed. Periaortic infrarenal LNs were removed in cases of positive periaortic inframesenterial LNs (frozen sections) or uterine corpus involvement detected during the preoperative workup. Not performing periaortic lymphadenectomy (neither inframesenteric nor infrarenal) was allowed in cases with small tumors and significant comorbidities. LN regions are presented in Figure 1.

As part of routine preoperative workup, all patients agreed to undergo preoperative gynecological examination (with a speculum, internal transvaginal and transrectal) (not in analgesia) and ultrasound imaging (transvaginal and/or transrectal and abdominal). Before the surgery, patients were also recommended to undergo additional pelvic (and abdominal) MRI, although this was not explicitly required in the study protocol. Exclusion criteria were as follows: neuroendocrine differentiation and preoperative FIGO stage IA or >IIA, distant metastases except in para-aortic LNs, scleroderma, lupus erythematodes, mixed connective tissue disease, secondary malignancy, previous radiotherapy of the pelvis, refusal to undergo preoperative imaging with ultrasound, underwent procedures other than TMMR and tLND, or did not undergo any surgery.

Potentially eligible participants were identified among those referred to the Department of Gynecological Oncology, Gdynia Oncology Center, Pomeranian Hospitals, Poland, who were suspected for or diagnosed (by biopsy) with cervical cancer, and who after initial examination fulfilled criteria for surgical treatment. Those referred with a suspicion of cervical cancer (on clinical examination, based on abnormal screening tests) underwent an initial biopsy or excisional procedure to establish a histological diagnosis of cervical cancer. The patients in this study represent a consecutive series of participants who fulfilled the inclusion and exclusion criteria.

### 2.3. Index Tests

Ultrasonography and MRI were performed by different experts who were blinded to the results of the other method and the final histology report (unavailable at the time of examination). Ultrasonography was performed by M.S., a gynecological oncologist with additional training in gynecological oncology ultrasonography. However, because surgical treatment is his primary line of work, M.S. was rated as a level 2 sonologist according to the European Federation of Societies for Ultrasound in Medicine and Biology classification [22].

Parameters for the assessment of pelvic tumor extent (local staging) were prospectively evaluated by each method and included:
Primary tumor detectable (yes/no).Primary tumor size (length, depth, width).Infiltration of parametria (yes/no).Infiltration of the uterine corpus (yes/no).Mesometrial (located in parametria) lymph node detection (yes/no).Infiltration of vagina wall/fornix (yes/no).Status of the anterior compartment (anterior to the cervix, border of the cervix and urinary bladder, and the urinary bladder).Status of the posterior compartment (posterior to the cervix, space between cervix and rectum, and the rectum).Suggested clinical stage (FIGO-2018, T from TNM (Tumor, Nodes, Metastases) system, and oT).Other (if noted; description).

#### 2.3.1. Endovaginal/Transrectal Ultrasonography

Ultrasonography for the assessment of pelvic tumor extent was performed using a Philips HD15^®^ ultrasound device (Philips Healthcare, Best, The Netherlands) with a 5–7 MHz endovaginal probe with a linear field of view. The probe was inserted transrectally (TRUS), or transvaginally in those who refused TRUS.

Tumor delineation was performed using a combination of gray-scale examination and power Doppler imaging. For this purpose, a color score was used according to definitions provided elsewhere [23] to distinguish the tumor from adjacent tissues based on neovascularization. The tumor was identified as an area of heterogeneous echogenicity that was different from normal cervical stroma (if present) or uterine corpus myometrium and/or had irregular borders (Figure 2a). Disruption of the cervical canal was an additional marker for identifying tumor borders; however, this was not obligatory (possible detection of a normal cervical canal and tumor delineation in the cervical stroma are shown in Appendix A). If the tumor was detectable, the ultrasound window was divided into two pictures, and the tumor was displayed in two planes to measure the length, depth, and width (Figure 2b).

The uterine corpus and cervical canal internal os were identified to define the border between the cervix and the uterine corpus. If the internal os could not be identified and the tumor was well delineated or the os was visible, but the tumor extended from the cervix into the myometrium, uterine corpus involvement was confirmed (examples are shown in Figure 3).

The parametrium is a composite tissue consisting of multiple structures with distinct developmental origins, such as the visceral endopelvic fascia, bladder mesentery, bladder adventitia, vascular mesometrium/mesocolpium, mesureter, ligamentous mesometrium/mesocolpium, neurogenital fascia, and paracervix. During TMMR, the paracervix should be resected because it is part of the Müllerian compartment, and the vascular and ligamentous mesometria/mesocolpia should be resected because these structures may contain first-line LN metastases and lymphangitic and in-transit tumor foci. All other structures should be preserved. It is impossible to differentiate these structures during preoperative assessment; therefore, the term ‘parametrium’ was used when describing preoperative findings. The status of the lateral parametria was examined as follows: the probe was positioned in the transverse plane, the cervix was identified and scanned from bottom to top and vice versa, the uterine vessels were identified (using gray-scale and power Doppler imaging) on the side of the cervix, the probe was directed toward one side (into the lateral parametrium), and gentle probe pushing was performed on the area (dynamic examination [24]) to detect any rigid structures in the field of parametrium protruding from the cervix (a sign of infiltration (Appendix A); this procedure was performed on different levels and both sides of the cervix in the same manner. In addition, infiltration of the lateral parametrium was recognized by the irregular tumor extension of hypoechogenic prominences into the region (Figure 4).

With the same probe application as that used for the parametria assessment, the lateral paracolpium regions were scanned dynamically [24] to search for rigid vascularized structures at the level of the vaginal fornices and upper third of the vagina.

Regions anterior and posterior to the cervix were examined to search for any possible bladder or rectum involvement and to examine the anterior and posterior parametria. The examination procedure was as follows: the probe was placed in the sagittal plane, moved in the direction of the anterior vaginal fornix, gently pushed forward and backward, and additionally second examiner’s hand pushed onto the abdominal wall over the symphysis in the direction of the cervix while the probe simultaneously being placed in the direction of the anterior fornix. Both maneuvers were applied to detect any tumor spread into the anterior direction towards the urinary bladder. A negative sign was defined when all structures were movable against each other, whereas a positive was when there was no sliding sign between the cervix and urinary bladder (additional criteria were edema of the urinary bladder mucosa; Appendix A). The probe was then placed in the sagittal plane and placed in the direction of the posterior vaginal fornix. First, both sides were scanned, and the probe was then gently pushed back and forth (dynamic scanning) to scan the posterior area of the cervix and upper vagina. Sliding or movability between the cervix and rectum or between the vagina and pouch of Douglas as well as delineation of the smooth posterior wall of the cervix were considered negative for involvement (Appendix A). Standard examination of the uterine corpus and adnexa was also performed.

#### 2.3.2. Abdominal Ultrasonography for Regional Staging

The retroperitoneal region around the inguinal, external iliac, internal iliac (paravisceral and obturator), common iliac, presacral, periaortic inframesenterial, and periaortic infrarenal regions (Figure 1) were scanned for enlarged or suspected retroperitoneal LNs, according to definitions provided elsewhere [25]. Furthermore, the abdominal cavity was scanned systematically to search for any suspected focal lesions. Ultrasonography for the assessment of possible regional spread was performed using a Philips HD15^®^ ultrasound device (Philips Healthcare, Best, The Netherlands) with a 2.4–5 MHz convex transabdominal probe.

The examination started with the probe being placed over the inguinal region, transverse to the long axis of the femoral vessels. Scanning started approximately 15 cm below the inguinal ligament, and the probe was moved cranially. Next, the probe was twisted approximately 90° to visualize the femoral veins along their longitudinal axis, and scanning was performed by moving the probe on both sides of the vessels. The probe was then repositioned transverse to the femoral vessels below the inguinal ligament and moved cranially, crossing the ligament. The probe was moved along the external iliac, along the common iliac vessels cranially, and up to the aortic bifurcation. Scanning of the vessel region between the inguinal ligament and aorta bifurcation, with the probe placed transverse to the long axis of the vessels, was performed several times by moving the probe cranially and dorsally. Next, the probe was twisted approximately 90° into the longitudinal axis of the iliac vessels, and the regions on both sides of the vessels were examined.

In cases where vessels were difficult to identify in gray-scale, the power Doppler was introduced to facilitate examination. Laterally, the iliopsoas muscle, medial bowels, and ovaries were visualized. The probe was then placed transversely at the level of the aorta bifurcation and moved slightly cranially and then dorsally to visualize the aorta bifurcating to both common iliac arteries, followed by visualization of the left common iliac vein and the region below (i.e., the presacral region).

Next, the probe was placed below the xiphoid process transversely and tilted in oblique directions to visualize the pancreas and the aorta over the pancreas with the truncus celiacus. The region around the aorta and inferior vena cava (VCI) were scanned from below the pancreas to the aortic bifurcation. First, the probe was moved transverse to the vessels in the dorsal direction and then the cranial direction several times. Then, the probe was twisted approximately 90° to visualize the vessels along their longitudinal axis, and the probe was moved laterally to scan the regions between the aorta and VCI as well as the prevertebral regions to the left of the aorta and the right of the VCI.

Appendix A shows the scanning of the retroperitoneal pelvic and periaortic regions. An example of full local and regional staging using ultrasonography in a patient with locally advanced cervical cancer (involvement of the uterine corpus) is shown in Appendix A. Additionally, the abdominal cavity was systematically scanned to detect any focal lesions suspected for metastasis.

#### 2.3.3. MRI

MRI was performed using a 1.5T Siemens Magnetom Essenza scanner with a phased-array surface pelvic coil. A standardized MRI examination technique was used, which included T2-weighted sequences in sagittal and axial oblique planes (perpendicular to the long axis of the cervix) and axial T1-weighted sequences. To assess tumor size, location, and extension to the parametria, T2-weighted images in two orthogonal planes, sagittal and axial oblique sequences (perpendicular to the long axis of the cervix), or axial sequences were used. Axial T1-weighted imaging from the symphysis to the level below the inferior mesenteric artery was performed to detect pelvic and periaortic LNs. Consequent slices with a width of 5 mm and a 1-mm gap were acquired with a 512 × 256 matrix, four signal averages, and a 24- to 28-cm field of view. Additionally, several examiners used T2-weighted oblique sequences, parallel to the long axis of the cervix (coronal oblique plane), T2- and T1-weighted sequences in the coronal plane, T1-weighted sequences in the sagittal plane, or T2-weighted images with fat suppression in the sagittal and axial planes. Intravenous gadolinium contrast was administered at a dose of 1 mmol/kg (Omniscan, GE Healthcare, Oslo, Norway). The same regions as ultrasonography, except for the periaortic infrarenal and upper abdomen, were scanned with the MRI (only the pelvis up to the inferior mesenteric artery was scanned by MRI). Additionally, computed tomography was performed to image the upper abdomen and regions around the upper part of the abdominal aorta.

#### 2.3.4. Measurements and Stage Prediction

The two parameters of tumor measurement and disease stage prediction were defined as follows. The tumor was measured along three dimensions (length, depth, and width). For the calculations of the tumors, we defined the maximal size, mean size, volume (calculated from the available three measurements and not from imaging device software), and substitutive diameter (d_s_), which was calculated as a sphere diameter of the same volume as the volume of tumor (V_t_):
(1)$$d_{s}=\sqrt[{3}]{\frac{6V_{t}}{\pi }},$$

##### Measurements of the Primary Tumor

If only a biopsy was performed before treatment, direct size measurements from imaging were compared with the size of the histological (reference) specimen from the TMMR.

If the patient underwent conization before imaging, tumor size was compared between imaging measurements (ultrasound and MRI), searching the residual tumor and those from the final histological (reference) specimen from the TMMR (residual tumor).

If the patient underwent imaging before conization, final histological (reference) measurements were compared with the sum of the conization measurements and those of the residual tumor from the TMMR specimen.

Residual tumor size (of the TMMR specimen with previous conization) was described in the pathology report (reference), and in cases where no or only microscopic tumor was detected, these were labeled with measurements 0-0-0 mm.

##### Stage Prediction

The staging of the local tumor before surgery was based on all the available information: imaging, clinical examination, and histology (biopsy or conization specimen). The stage was predicted by the comparison of ultrasound and MRI data. In cases of discordance between imaging and clinical examinations, the highest stage was used regardless of whether it was based on the imaging or clinical examination. For treatment decisions, in cases of discordance between the clinical examination and MRI, the clinical examination and local staging by ultrasound were used.

For ultrasonography, clinical local stage was established based on ultrasonography and clinical examination, and in patients who underwent conization, the diameter of the tumor in the cone was added to that measured by ultrasound (i.e., residual tumor detected by imaging).

For MRI, clinical local stage was established based on the MRI and clinical examination, and in cases who underwent conization, the diameter of the tumor in the cone specimen was added to that measured by MRI (i.e., residual tumor as detected by imaging).

In cases where imaging (MRI or ultrasound) detected signs of involvement of the vaginal fornix/wall, but physical examination of the vagina did not, stage IIA (T2a) was assigned because the tumor may infiltrate the vaginal wall from the abdominal side (as suggested in imaging) without being detected by vaginal examination. However, if clinical examination revealed involvement of the vaginal fornix but imaging did not, stage IIA (T2a) was assigned.

Three available staging systems were used for preoperative evaluation and postoperative assessment: FIGO 2018 classifications [26]; TNM system by the Union for International Cancer Control, eighth edition, 2016 [1]; and oT staging (Appendix A) [27].

### 2.4. Reference Standard

The reference standard was a histological examination with immunohistochemical evaluation in cases with inconclusive hematoxylin and eosin testing of the surgically removed uterus with mesometria, vaginal fornices, and retroperitoneal LNs.

Definitions of and rationale for positivity and categories of the index test are described along with the methodology of examination. The definition of and rationale for test positivity and categories of the reference standard was as described elsewhere [27,28].

The reference standard (histology of the final specimen) results were not available to the ultrasound or MRI examiners. Only results of the biopsy (cervical cancer confirmation) and clinical/gynecological examination were available to the index test reader. None of the detailed information of the index tests was available to the assessors of the reference standard, except for the general clinical diagnosis of cervical cancer based on a previous biopsy.

This paper was written according to the standards for reporting of the diagnostic accuracy (STARD) initiative [29].

### 2.5. Statistical Analysis

First, the accuracy of the ultrasound was calculated against the reference standard in all included participants. Second, a comparison of the accuracy between ultrasound and MRI (against the reference standard) was performed in patients who had undergone both imaging examinations.

The confusion matrix was used to compare the binary parameters (i.e., Yes/No) of the imaging methods [30]. The confusion matrix is a specific table layout that allows visualization of the performance of the imaging method relative to histopathological examination (Table 1). Each column of the matrix represents instances of the reference class (histopathological examination), whereas each row represents the instances of the imaging method class (ultrasonography or MRI).

For each imaging method, five parameters were determined based on the confusion matrix using the following Formulas (2)–(6):
sensitivity, true-positive rate (TPR):
(2)$$TPR=\frac{TP}{TP+FN},$$specificity, true-negative rate (TNR):
(3)$$TNR=\frac{TN}{TN+FP},$$precision, positive predictive value (PPV):
(4)$$PPV=\frac{TP}{TP+FP},$$negative predictive value (NPV):
(5)$$NPV=\frac{TN}{TN+FN},$$accuracy (ACC):
$$ACC=\frac{TP+TN}{TP+FP+TN+FN},$$

In cases where there were more classes for determining tumor stage, only the accuracy of a given imaging method relative to that of the histopathological examination was determined. Continuous data are presented in medians, ranges, or inter-quartile ranges (IQR), while categorical data are given as percentages. Confidence intervals (CI), when applicable, are given for the 95% range. The accuracy of both MRI and ultrasound was compared using the exact McNemar’s test. To determine whether the differences in test performance between MRI and ultrasound were significant, McNemar’s test for paired data was applied to calculate p-values. The differences in the outcome of both methods were considered statistically significant with a *p*-value of 0.05 or less. Data analyses and other calculations were performed in RStudio Desktop (version 1.1.463) software [31] using R [32]. There were no indeterminate index test or reference standard results.

Participants with missing ultrasound examinations were excluded from the study. Patients with missing MRI examinations were excluded from the comparison of ultrasound and MRI. There were no missing reference test data among patients who were included in the study and underwent the TMMR/tLND procedure.

Variability analysis was performed for the diagnostic accuracy of ultrasound and MRI depending on BMI and tumor histology and grade, and for ultrasound, variability analysis was performed for the route of probe insertion: transvaginal or transrectal.

The intended sample size was not calculated for this study because it was an interim sub-analysis of the TMMR-RS trial, which had different endpoints.

## 3. Results

Data from consecutive participants participating in the TMMR-RS, from Gdynia Oncology Center, Pomeranian hospitals, who underwent surgical treatment between January 2017 and April 2021 were evaluated. The recruitment flow of participants is presented in Figure 5. Baseline demographic and clinical characteristics and distribution of disease severity of participants are presented in Table 2. Imaging data are available in Table 3. All patients underwent TMMR and tLND via laparotomy. None of the patients received adjuvant radiotherapy. Adjuvant chemotherapy was applied after consultation with a medical oncologist, mainly for patients with metastatic LNs.

The median time interval between imaging (index test) and surgery (reference standard) was 20 days (range 1–30 days). No clinical interventions, other than computed tomography (if indicated), were performed in patients between the index and reference indices.

### 3.1. Tumor Detection and Tumor Size

Accuracy of tumor detection and localization using ultrasonography are shown in Table 4.

Comparison of ultrasonography and MRI performance for tumor detection and localization are shown in Table 5. In a tested cohort (*n* = 50), there were no significant differences between these two imaging methods.

Primary or residual (after conization) cervical tumor measurement errors between imaging and histology are shown in Table 6 and Appendix A.

### 3.2. Parametria

The accuracy of ultrasound and MRI for parametria assessment are shown in Table 4 and Table 5. With ultrasonography, parametrial involvement was accurately excluded in every patient with histologically negative parametria, whereas false-positive findings were noted for MRI assessments.

Of the five patients with lateral parametria involvement based on histology, none were suspected by ultrasound (all examined using the transrectal approach) and one was suspected by MRI (four of the five patients had MRI data available). Of these five patients with histologically positive parametria, involvement was proximal only (minimal, just at the border of the cervix, with the middle and distal parts of parametria negative for the disease). However, one patient had unilateral lateral parametrium involvement occupying the proximal and middle parts, with a negative distal part and resection margins. This patient was preoperatively staged as T2a2 according to both MRI and ultrasound, which showed negative parametria involvement.

Lateral parametria involvement was suspected in preoperative ultrasonography in none of the patients (0/58) who underwent TMMR/tLND. As for MRI, there were 10 patients with positive lateral parametria in preoperative MRI (10/50) who underwent TMMR/tLND. Correct assignment (confirmed by the pathology report) was made in 1 out of 10 patients (false positive rate: 19.6%).

### 3.3. Uterine Corpus Involvement

The accuracy of ultrasound and MRI for uterine corpus assessment are shown in Table 4 and Table 5. Of the patients who were preoperatively suspected of uterine corpus involvement (tumor extending over the internal os) by ultrasonography (*n* = 5), four had uterine corpus involvement confirmed by histology, and all but one (because of obesity and comorbidities) underwent periaortic infrarenal lymphadenectomy. Of the three patients with histologically confirmed uterine corpus involvement and infrarenal periaortic lymphadenectomy, one had metastatic LNs in the infrarenal and inframesenteric regions, without the involvement of any pelvic LNs. The other two patients had negative periaortic LNs, a single positive LN in the paravisceral region, and mesometrial LNs. The patient in whom ultrasonography suggested uterine corpus involvement, which was confirmed by histology, but who did not undergo infrarenal periaortic lymphadenectomy because of abdominal obesity and comorbidities (and the MRI was negative for uterine corpus involvement), positive LNs were found in the mesometrial, external iliac, and paravisceral regions, whereas other higher-level LNs, including common iliac, presacral, and periaortic inframesenterial regions, were negative. However, the patient recurred in the infrarenal periaortic region with involvement of the duodenum 16 months after the surgery (no local recurrence in the field of the previous TMMR/tLND).

There were 7/50 patients suspected of uterine corpus involvement based on MRI. Among these, three had uterine corpus involvement based on the final pathology report.

### 3.4. Mesometrial LNs

We suspected an enlarged mesometrial LN on one side in the ultrasonography of one patient. The MRI suggested unilateral parametrium involvement and not the LN. The final histology revealed proximal, bilateral parametrial involvement, with negative middle and distal parts of the parametria and clear surgical margins. Of the seven patients who had metastatic mesometrial LNs, none had enlarged suspected LNs in this region in neither the ultrasonography nor MRI.

### 3.5. Vaginal Wall/Fornix Infiltration

The accuracy of the ultrasound and MRI for vaginal wall/fornix assessment are shown in Table 4 and Table 5. We suspected vaginal fornix involvement on the left side in one patient, based on ultrasonography and clinical examination. The same was revealed by MRI. The final histology report revealed parametrial involvement on the left side and not the vaginal wall/fornix. Based on MRI, 13 out of 50 (26%) patients had involvement of the vaginal fornix/wall, and one was confirmed in the final histology (false positive rate: 24.5%). This location was positive in two cases based on histology; neither was suspected by ultrasonography, and one was suspected by MRI.

### 3.6. Regions Anterior and Posterior to the Cervix

There were no patients in whom involvement of the regions anteriorly or posteriorly to the cervix was suspected by ultrasonography, and none of the patients showed findings in final histology reports. There were cases where the tumor involved the majority of or all of the cervix; however, the tumor did not cross the cervix compartment border anteriorly or posteriorly.

There were no patients in whom involvement of the region anterior to the cervix was suspected by MRI; however, two cases were suspected for infiltration from the cervix posteriorly to the rectum (false-positive rate: 4%). This was not confirmed by the final histology report.

### 3.7. Stage Prediction

Preoperative accuracy of ultrasonography and MRI for cervical cancer local staging according to the three different staging systems are shown in Table 7 and Appendix A. Ultrasonography was significantly more accurate than MRI in predicting the correct local stage of cervical cancer.

The variability analysis for the diagnostic accuracy of ultrasound and MRI was not performed because there was insufficient data for stratification. Moreover, because ultrasonography was performed by a single examiner, the variability of the examiner’s performance could not be calculated.

### 3.8. Regional LNs

The status of pelvic and para-aortic LNs was not assessed in the final statistical analysis because of the small number of patients with enlarged suspected LNs detected by preoperative ultrasonography and MRI. Nevertheless, the available results are described below.

The preoperative abdominal ultrasonography showed one patient with a suspected enlarged LN located in the paravisceral region, which was also detected by MRI. The patient was a 23-year-old woman who, after consultations with a gynecological oncologist, radiotherapist, and medical oncologist, decided to undergo TMMR/tLND. Metastasis was confirmed in the one paravisceral LN without capsule extension. Additional metastases were diagnosed in small intercalated mesometrial LNs on both sides. All other LNs (a total of 74) were negative.

The preoperative MRI showed five patients who were suspected to have metastatic external iliac LNs, although the final histology report confirmed nodal involvement in one case only. However, this patient had a metastatic LN on the opposite side than what was suggested by the MRI. Thus, the positive rate of MRI-suspected LNs was 0% for this patient.

The preoperative MRI revealed one patient with suspected metastases in paravisceral LNs on both sides. The final histology report confirmed a positive node on one side. This was the 23-year-old woman described earlier.

There were no patients who had suspected LNs in the common iliac, presacral, or paraaortic regions in the preoperative MRI.

There were 11 patients with positive external iliac LNs based on histology reports. Among these, one patient had a suspected metastatic LN in this region in the MRI (but not in the ultrasound); however, the side did not match: MRI showed that the right side was positive, whereas histology showed that the left side was positive. No other patients had suspected LNs based on preoperative imaging (all underwent ultrasound, and all but two underwent MRI).

Eight patients had positive paravisceral LNs based on final histology reports, which was suspected by imaging in only one 23-year-old patient. No other patients had suspected LNs based on preoperative imaging (all underwent ultrasonography, and all but one underwent MRI).

There were two patients with positive common iliac LNs based on final histology reports. Neither was suspected by imaging (both underwent ultrasound, and one underwent MRI).

There were two patients with positive presacral LNs based on final histology reports. Neither was suspected by imaging (both underwent ultrasound, and one underwent MRI).

Two patients had positive periaortic inframesenterial LNs based on final histology reports. Neither was suspected by imaging (both underwent ultrasound and MRI).

One patient had positive para-aortic infrarenal LNs based on the final histology report, which was not suspected by imaging (patients underwent ultrasound and pelvic/abdominal MRI).

There were no adverse events from performing the index or reference standard tests. However, one patient could not tolerate MRI because of claustrophobia.

## 4. Discussion

Accurate local and regional tumor staging using ultrasound is feasible in patients with cervical cancer. Accuracy of tumor detection and size measurement were comparable between ultrasound and MRI. We could almost perfectly predict the absence of parametrial, uterine corpus, or vaginal wall involvement (specificity 98–100%) using ultrasonography. In general, the accuracy of ultrasonography was not different from MRI for assessment of parametrial, uterine corpus or vaginal fornix involvement. Patients were more accurately assigned to the correct disease local stage by ultrasound than by MRI.

### 4.1. Tumor Detection and Tumor Size and Volume

Primary and residual tumors can be well delineated using both ultrasonography and MRI. The accuracy of tumor size measurements was largely comparable between ultrasonography and MRI. However, a smaller error was observed for ultrasonography than for MRI, relative to histology, and this was not associated with the type of measurement.

Tumor size is not important if cancer field surgery is applied because provided the tumor grows within the compartment, the appropriate extent of surgery is applied [11]. For this reason, we did not analyze the degree of cervical stroma infiltration (less or more than two-thirds).

In studies where histological examinations of the uterus specimen after surgery were used as the reference index, sensitivity of ultrasonography and MRI for the tumor identification was 90–93% and 67–88%, respectively [2,6,33], and specificity was 95–97% and 84–89%, respectively [2,6]. Detection of the tumor using different imaging modalities is dependent on the size of the tumor. The agreement between ultrasound and pathology was excellent for correctly classifying bulky tumors (>4 cm) and good for classifying small tumors (<2 cm). The agreement between MRI and histology was good for classifying tumors < 2 cm or >4 cm [6]. In a subgroup of small tumors (≤1 cm^3^), ultrasonography was better than MRI for tumor detection, which was shown by a sensitivity of 72% and 44%, respectively [2].

The diameter or volume of the primary tumor may be of significance if a more classical approach is implemented; that is, if surgery would be applicable for patients with a FIGO-2018 stage up to IB2, whereas patients with stage IB3 would be referred for definitive chemoradiotherapy. Craniocaudal tumor diameter is considered important because this measurement cannot be obtained by means other than imaging. In a study by Testa et al., the mean difference between histopathological and ultrasound measurements of the craniocaudal diameter of the tumor was 0.62 mm, and the mean difference between histopathological and MRI measurements of the craniocaudal diameter of the tumor was 1.49 mm [33]. Moreover, in a cohort of 182 patients, maximal tumor diameter did not differ significantly when measured by ultrasound (27.6 mm (standard deviation (SD) ± 16.0)), MRI (28.6 mm (SD ± 15.6)), or histology (26.6 mm (SD ± 16.9)) [6]. Once the tumor volume is automatically calculated after estimating its three diameters by applying the formula for an ellipsoid [34], ultrasound volumetry correlated more precisely with pathology results than did MRI volumetry [2].

### 4.2. Parametria

One of the goals of the diagnostic evaluation for TMMR and tLND is to exclude parametrial involvement. True-negative findings were assessed perfectly by ultrasonography (specificity of 100%), whereas MRI produced some false-positive findings (specificity of 80%). It is noteworthy that none of the patients had parametrial involvement based on preoperative ultrasonography. Furthermore, all patients with false-negative findings underwent ultrasonography with the transrectal approach, which may be an important issue because gynecologists are trained in and primarily perform transvaginal examinations in routine practice. Moreover, most practitioners are not trained to examine parametria during ultrasonography training. The transrectal approach is used specifically in patients with cervical cancer because it is considered more accurate for local assessment of the disease [2].

If patients had positive parametria based on imaging, they were staged as T2b and were therefore not included in the TMMR-RS, which may introduce some bias to the results. However, we had surgical and pathological references from patients who underwent surgery. Moreover, it would not be possible to obtain such a strong reference index from patients referred for definitive chemoradiotherapy.

According to the results of Hockel et al. [11] and the cancer field theory [35,36], as long as the tumor is within one compartment and surgical margins are negative, the addition of adjuvant radiation provides no benefit for overall survival. Furthermore, the risk of complications from combined treatment is much higher. In our study, all patients with pT2b had negative surgical margins. Moreover, in the four of five patients classified as pT2b, infiltration was observed only at the border of the cervix and paracervix, not palpable even during surgery, and only detected in the histological examination. One patient had paracervix involvement to a greater extent, but the surgical margins were negative. Detailed information on general recommendations and cancer field surgery concepts were provided to each patient, and consultations with radiation therapy consultants and medical oncologists were offered. None of the patients desired adjuvant radiation; however, most underwent adjuvant chemotherapy. In our group, four out of five patients with positive parametria had a follow-up of 12–36 months, and none recurred during the follow-up period, despite not receiving adjuvant radiation.

In studies that carried out histological examinations of uterus specimens after surgery as a reference index, sensitivity of ultrasonography and MRI for the detection of parametrial involvement was 60–83% and 40–69%, respectively [2,6,33], and specificity was 98–100% and 92–98%, respectively [2,6]. However, it is worth acknowledging that in each study, there was a small number of patients with positive parametria based on histological examinations: 6/95 (6.3%) [2], 5/68 (7.4%) [33], and 13/182 (7.1%) [6]. Thus, any imaging method may be limited in its ability to identify minimal tumor invasion into the parametria [6]. In another study, a small subset of patients (*n* = 25) who underwent ultrasonography for preoperative local staging showed 100% concordance between ultrasound and histology after surgery for parametrial involvement. However, the mean maximal cervical lesion diameter was 1.6 cm, and the median age of patients was 44 years [37]. Thus, this cohort represents a highly selective set of candidates for surgical treatment in which parametrial involvement would be extremely rare. In another study [38] in a cohort of 400 patients, of whom 41% had parametrial invasion confirmed by pathology reports, the accuracy of MRI for detecting parametrial involvement was compared with clinical examination under general anesthesia. The accuracy of the latter, if performed by a gynecological oncologist (who had access to the MRI), was higher than that of the former. Moreover, the accuracy of examination under anesthesia was not affected by tumor size, whereas the accuracy of MRI decreased if tumors were larger than 2.5 cm [38].

Two recent meta-analyses concluded that ultrasound and MRI have similar diagnostic performance for assessing parametrial infiltration in cervical cancer, with a pooled sensitivity of 67–78% and specificity of 94–96% [39,40]. However, because of the heterogeneity of studies included in the meta-analyses and the relatively small sample sizes, they noted that the findings must be interpreted with caution and suggested further well-designed studies [39]. The best reference standard for comparing MRI and ultrasound assessments of parametrial invasion in cancer of the uterine cervix is histological examination of the whole parametria. This can be obtained once the patient undergoes surgical treatment. It is expected that most, if not all, patients will have negative parametria after surgery based on histology because every case suspected of parametria involvement during the pretreatment workup should be offered definitive radiochemotherapy [1]. Thus, there would be insufficient cases with positive parametria based on imaging, which biases the results of every study in patients with early cervical cancer who are eligible for surgical treatment. It may be easier to conduct such studies in patients undergoing cancer field surgery (TMMR or extended mesometrial resection (EMMR) plus tLND) for the primary treatment of cervical cancer, where patients with suspected parametrial involvement can still undergo surgical treatment but also provide specimens for histological analysis (i.e., the reference standard).

### 4.3. Uterine Corpus Involvement

Uterine corpus involvement was not evaluated in previous studies comparing ultrasound and MRI accuracy for staging cervical cancer [2,6,33,37]. Moreover, it has not been incorporated into the FIGO [26] or TNM [1] staging classifications. However, it is worth noting that once the tumor invades the uterine corpus, an additional lymph drainage route may be affected, which is not only via uterine vessels but also ovarian vessels directly to periaortic regions, similarly to endometrial cancer [41,42].

This was observed directly in one of our patients in the study and indirectly in another: recurrence in the periaortic infrarenal region with uterine corpus involvement at the primary surgery, and negative common iliac, presacral, and periaortic inframesenteric LNs. In addition, Hockel et al. reported patients with uterine corpus involvement (*n* = 109) [11], and in eight (89%) of nine patients with perimesenteric LN metastases, cervical cancer infiltrated the uterine corpus or isthmus. Periaortic LN metastases without metastatic occupation of pelvic second-line LN regions and infiltration of the uterine isthmus or corpus prognostically equated to pelvic first-line metastases [11]. Furthermore, they found a different prognostic value for periaortic LN metastases based on whether or not second-line basins contained metastases. All patients whose basins did not contain second-line metastases had local tumor infiltration of the uterine corpus, or at least of the isthmus. Survival of these patients did not differ from that of patients with first-line LN metastases of the pelvis, whereas periaortic LN metastases in the presence of second-line metastases were associated with poorer prognoses [11]. Involvement of the uterine corpus upstages the disease according to the oT staging system from oT1 to oT2. Patients with oT2 cancer have a poorer prognosis than those with oT1 cancer; however, the difference is smaller than between oT2 and oT3 [11].

### 4.4. Mesometrial LNs

Mesometrial LNs are small nodes placed along the route between the cervix and first-line pelvic (external iliac and paravisceral (obturator)) LNs. Thus, the detection of metastatic nodes using available imaging techniques (e.g., high-quality ultrasonography and MRI) is challenging, as was demonstrated in our cohort. Moreover, it would be technically extremely difficult to differentiate metastatic LNs in the mesometrium from actual parametrial infiltration. Nevertheless, mesometrial LNs are considered first-line nodes for cervical cancer, similarly to external iliac and paravisceral nodes [11].

### 4.5. Vaginal Wall/Fornix Infiltration

Similar to uterine corpus involvement, vaginal fornix/wall infiltration progresses the disease to stage oT2; thus, prognosis in these cases is poorer than those without extracervical disease [11]. According to general guidelines, patients at stage T2a1 are still considered candidates for radical surgery and systematic pelvic lymphadenectomy, and radical surgery is an alternative option for patients at stage T2a2. However, the quality of surgery is of key importance [1].

There were only two patients with histologically confirmed vaginal fornix involvement; thus, no conclusions could be made. Nevertheless, MRI seemed to be an inaccurate imaging method for this localization because a quarter of patients had suspected vaginal fornix involvement, which was confirmed in only one patient. Precise preoperative assessment and defining vaginal fornix involvement is important for planning an extension of the surgical margins within the Müllerian vagina (the human vagina has dual ontogenesis; the proximal part is derived from the Müllerian system, and the distal part is derived from the urogenital sinus) [11].

One study compared ultrasound and MRI for the detection of vaginal wall infiltration in only three patients who were positive based on histology. It was suspected by ultrasonography in one patient and by MRI in none of the participants [33].

### 4.6. Regions Anterior and Posterior to the Cervix

We chose to analyze imaging accuracy for detecting the involvement of regions anterior and posterior to the cervix because we aimed to search the tumor in every direction, and not only the lateral parametria. Similar to lateral parametria involvement, patients suspected to have disease spread in the urinary bladder or rectum would not be eligible for the TMMR-RS, which may bias the results. Nevertheless, we evaluated this localization using ultrasonography in every patient before surgery in the same way it was evaluated using MRI. MRI is regarded as the gold standard for the local (and regional) staging of cervical cancer. However, this has led many clinicians to disregard clinical examinations and base treatment decisions on MRI only, especially when locally advanced disease is noted by the radiologist. This was the case in two of our patients, who had suspected rectal involvement in their MRIs and were thus referred for definitive chemoradiotherapy. The radiotherapist examined patients and found discordance between the MRI and clinical examinations. We then performed ultrasonography, which showed a clear sliding sign between the cervix and rectum. These patients underwent surgery, and the space posterior to the cervix was free of disease.

Testa et al. evaluated ultrasound and MRI for the detection of vesicovaginal and rectovaginal septa infiltration [33]. Similar to our study, patients with apparent involvement of the bladder or rectum were excluded from the study, and surgical–pathological examination was the reference standard. Ultrasound examination correctly identified the only case of infiltration of the vesicovaginal septum and accurately diagnosed the absence of rectovaginal septum infiltration in all cases [33]. MRI examination yielded two false-positive diagnoses of infiltration of the vesicovaginal septum and one false-positive diagnosis of cancer infiltration of the rectovaginal septum [33]. The single case of infiltration of the vesicovaginal septum was characterized by disruption of the endopelvic fascia, as documented in the histopathological examination [33].

### 4.7. Stage Prediction

Pretreatment stage prediction is carried out using a combination of all available parameters from clinical and imaging examinations. Both local and regional staging must be performed to plan optimal treatment for each patient. We showed that the more complex the staging system was, the less accurate was the imaging. The most precise was ultrasonographical stage assignment for oT staging, with stratification into two stages only. The least accurate was MRI for the FIGO 2018 staging system.

In a cohort of patients who underwent cancer field surgery (TMMR or EMMR and tLND) for cervical cancer, treatment was based on clinical oT staging obtained during examination under analgesia, with the patient’s pelvic MRI scans available in the operating room. MRI findings were used to plan the surgery (extent of laparotomy and selection of surgical instruments), but not to exclude patients with cervical cancer from surgical treatment. They found that disease-specific survival was better stratified by oT staging than by FIGO or pT staging [11].

For classical approaches, the details of local staging are important because patients with a tumor larger than 4 cm and parametrial involvement will be assigned to stages IB3 and IIB, respectively, and the treatment plan for both would likely be definitive radiochemotherapy. Moreover, patients with bulky regional LNs are candidates for non-surgical treatment [1].

### 4.8. Regional LNs

To the best of our knowledge, there has not been any study that has systematically evaluated the accuracy of ultrasonography for regional LN status in patients with cervical cancer. However, there have been reports on the use of transabdominal ultrasonography to detect ureteral obstruction [6,33] or assess periaortic LNs [33]. In one study [33], the authors reported the accuracy of ultrasound and MRI for the detection of metastatic pelvic LNs. There were 11 patients with positive LNs, of which one was suspected by presurgical ultrasound, whereas three were suspected by MRI. However, LN status was not the aim of the study, and specific details of the ultrasound examination methodology were not described [33]. In another study that compared ultrasound and MRI for tumor delineation in cervical cancer (not regional LNs), 38 (20% of all undergoing surgery) patients had positive LNs based on the histological workup, of which three were suspected by ultrasound and four by MRI, and most women with LN metastasis and micrometastasis had seemingly normal-sized LNs [6].

Using a standard ultrasound machine, enlarged LNs can be detected, which leaves all normal-sized LNs as non-suspicious. Enlarged LNs are not always metastatic, and LNs of normal size can carry metastases, which is the most frequent scenario in patients with cervical cancer. Contrast-enhanced ultrasonography may help solve this issue. Furthermore, the addition of positron emission tomography (PET) to ultrasonography (performed for local staging) [37] is another possible solution to this problem. However, PET is not widely available and is cost-intensive. Moreover, in a recent meta-analysis, MRI, computed tomography, and PET performed comparably for assessing nodal metastases, with low sensitivity (29–69%) but high specificity (88–98%), even when stratified for anatomical location (pelvic or para-aortic) and level of analysis (per patient vs. per site) [40].

Lymph drainage from the uterine cervix and the corpus has been described previously [43,44]. Additional insight has been gained using tracers and robotic surgery to follow lymph drainage and identify the compartments corresponding to the primary cancer site, the cervix [41,45], and the uterine corpus [42,46,47]. Thus, provided there are no bulky, difficult-to-remove LNs, the localization of possible metastases is predictable in most cases of cervical cancer, and the extent of the surgery can be planned adequately. Therefore, failing to detect normal-sized and -shaped metastatic LNs using imaging is largely unproblematic. Nevertheless, whether LN extracapsular extension can be predicted by imaging preoperatively is worth investigating because this factor is of prognostic significance [48].

Our study has several limitations. It was conducted in a single institution, and ultrasonography was performed by a single examiner on one ultrasound machine; thus, the results cannot be generalized. However, the study was conducted in a regular gynecologic oncology department and examination was performed by an experienced gynecologist oncologist, and thus, under the above-mentioned criteria, the results of our study could be applied to this setting. The number of included patients was low, although we performed interim analyses of various endpoints of the TMMR-RS (initial oncological results of the trial are currently under review). Two different insertion methods of the endoprobe may influence the final results of the ultrasound examinations. MRIs were evaluated by different experts, and one who specializes in gynecological oncology may perform assessments differently. In addition, all patients included in the trial underwent surgical treatment of cervical cancer, and patients with more advanced metastatic disease (e.g., apparently enlarged LNs, involvement of parametria, or involvement of regions anterior and posterior to the cervix) were not included. Nevertheless, histology examinations of specimens after surgery provided true and verifiable extents of local and regional disease as reference standards, which would be impossible to obtain from irradiated patients. Several studies evaluated the accuracy of imaging in patients across all stages of cervical cancer [5,49,50], and such a patient cohort including those with apparent parametrial involvement, bladder and rectum infiltration, and metastatic LNs would provide sufficient data to detect positive imaging findings. However, ultrasound findings would then be compared with MRI findings, which is considered the gold standard, and histology would not be available as a reference standard; thus, the true accuracy of MRI would be questionable, as described earlier.

If cancer field surgery is applied to the treatment of cervical cancer, then the role of pretreatment imaging differs from that in the classical approach. Although regional staging of LNs is important, provided there are no large bulky LNs in the imaging that would be technically challenging to remove, and cancer field surgery is applied, preoperative evaluation of the retroperitoneal space is of secondary importance. Moreover, given that cervical cancer is a disease that follows the “order of cancer” [35] for a prolonged period during its development, local staging is most important. Based on a few local parameters, which must be evaluated during the preoperative workup, the extent of the surgery can be planned. In cases of apparent parametrial involvement, EMMR should be planned [51], whereas in cases of uterine corpus involvement, periaortic, perimesenteric, and infrarenal lymphadenectomy should be performed because first-line LNs may be involved (an example is shown in Appendix A). In cases of vaginal wall/fornix involvement, the length of the vaginal resection should be extended, which can be performed according to MRI and an examination under general anesthesia (in apparent locally advanced disease), as described previously [10,20,38]. However, given the high accuracy, availability, and improved quality, ultrasonography may be considered the tool of choice. Although, it must be stressed that examinations should be performed by an experienced operator because of the considerable difference in performance between beginners and experienced examiners [52]. Our study presents the accuracy of ultrasonography performed by an experienced gynecological oncologist with additional training in sonology. Furthermore, the extent of surgery must be adapted according to intraoperative clinical and histological findings (i.e., frozen sections of removed LNs).

## 5. Conclusions

For patients with cervical cancer who are eligible for surgical treatment with TMMR and tLND, ultrasonography performed by a gynecological oncologist provides high accuracy for local staging of the disease and is thus useful for planning adequate surgical treatment. Its accuracy was not inferior to that of MRI interpreted by a radiologist for tumor detection and the assessment of parametrial, uterine corpus and vaginal fornix involvement. Ultrasonography was superior to MRI in assigning patients to a correct disease stage. Regional staging with ultrasonography is feasible; however, it requires evaluation in further well-designed studies.

The future of imaging for the pretreatment staging of cervical cancer may develop in two directions: fusion of different modalities, as was shown in a recent study [53], or further improvement of the technology of each method and examiners’ expertise. A more precise study on parametrial involvement is needed and could be performed as a sub-analysis within clinical trials on TMMR and EMMR.

## Figures and Tables

**Figure 1 diagnostics-11-01749-f001:**
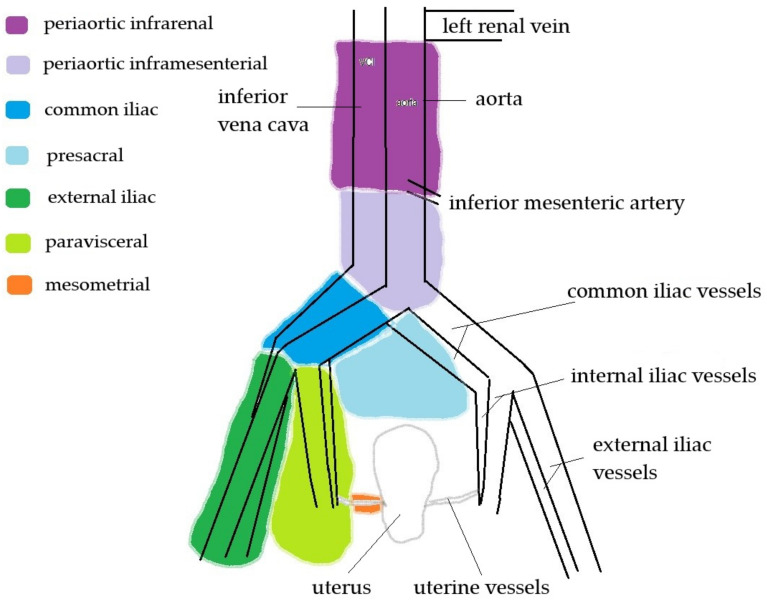
Schema of lymph node basins.

**Figure 2 diagnostics-11-01749-f002:**
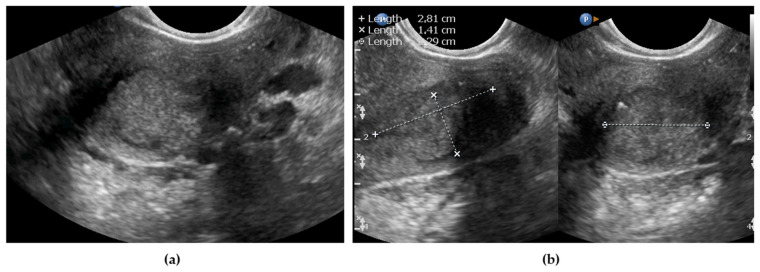
Cervical cancer primary tumor imaged using ultrasonography: (**a**) delineation of the tumor; (**b**) tumor measurements (lines indicate calipers on the tumor).

**Figure 3 diagnostics-11-01749-f003:**
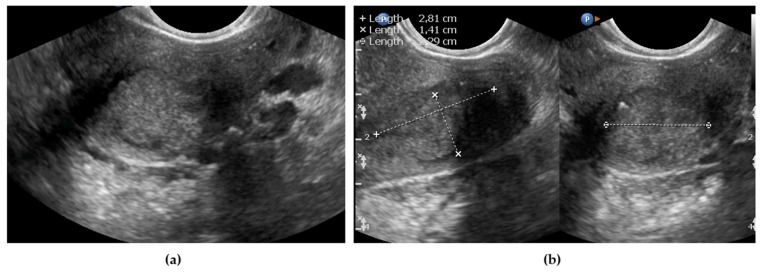
Tumor of the cervix in relation to the uterine corpus imaged using ultrasonography: (**a**) tumor does not involve the uterine corpus (calipers for tumor measurement); (**b**) tumor involves the uterine corpus (yellow arrows delineate the cervical tumor with involvement of the anterior wall of uterine corpus; blue arrows indicate the normal endometrium, lines indicate calipers on the tumor).

**Figure 4 diagnostics-11-01749-f004:**
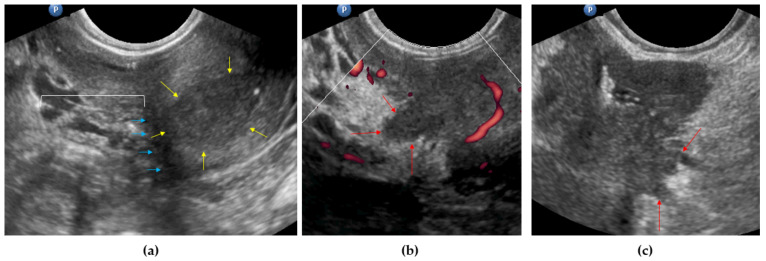
Lateral parametria assessment imaged using ultrasonography: (**a**) parametrium not involved (yellow arrows delineate the cervical tumor; blue arrows delineate the right lateral border of the cervix (not involved by the tumor); the white line represents the area of the lateral parametrium); (**b**) right parametrium involved (red arrows indicate the tumor prominence into the parametrium fat); (**c**) left lateral parametrium involved (red arrows indicate the tumor prominences into the parametrium fat).

**Figure 5 diagnostics-11-01749-f005:**
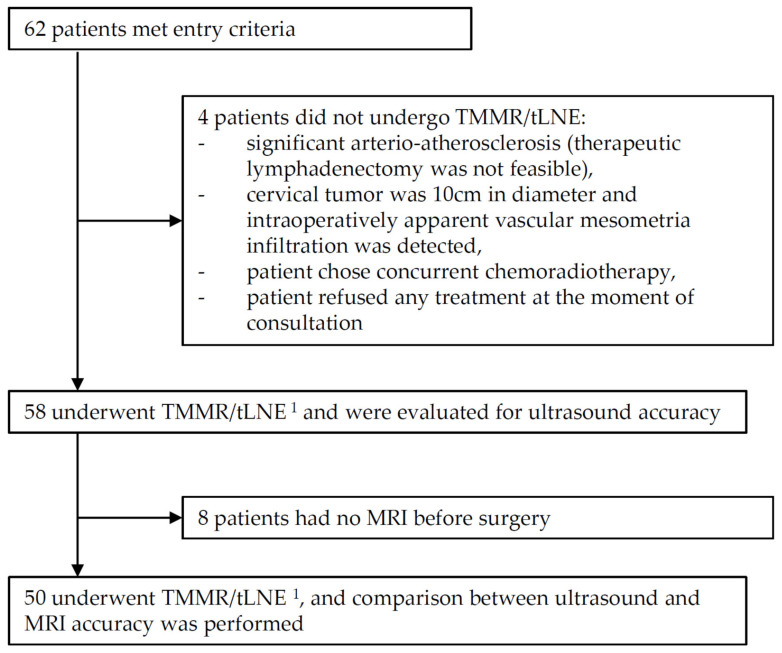
The flow of patients; ^1^ TMMR/tLND, total mesometrial resection, and therapeutic lymphadenectomy.

**Table 1 diagnostics-11-01749-t001:** Confusion matrix template.

Comparison Parameter	Histological Examination		
Ultrasonography or MRI		Yes	No
	Yes	True positive (TP)	False positive (FP)
	No	False negative (FN)	True negative (TN)

**Table 2 diagnostics-11-01749-t002:** Patients and disease characteristics (*n* = 58).

Clinical Parameters	Data
Age, years, median (range)	48.5 (23–80)
BMI ^1^, kg/m^2^, median (IQR)	26.6 (6.65)
Previous conization	11 (19%)
Histological type	
Squamous cell carcinoma	47 (81%)
Adenocarcinoma	10 (17%)
Adenosquamous carcinoma	1 (2%)
Histological grade	
G1	10 (17%)
G2	38 (66%)
G3	10 (17%)
LVSI ^2^ status	
Positive (+)	17 (29%)
Negative (−)	41 (71%)
Neoadjuvant chemotherapy	0
Therapeutic lymph node dissection	
External iliac, paravisceral, common iliac and presacral	58 (100%)
Periaortic inframesenteric	56 (97%)
Periaortic infrarenal	9 (16%)
Number of LNs ^3^ removed per patient, median (range)	57 (32–130)
Maximal tumor size, in histology ^4^, mm, median (IQR)	32 (22.25)
pT stage, from TNM, 8th ed., 2016	
pT1a	1 (2%)
pT1b1 (< 4 cm)	38 (65%)
pT1b2 (> 4 cm)	14 (24%)
pT2a1	0
pT2a2	1 (2%)
pT2b	5 (8%)
pN stage, from TNM, 8th ed., 2016	
pN0	38 (66%)
pN1	20 (34%)
Numer of metastatic LNs (including mesometrial) per patient, median (range)	1 (1–7)
Pathological ontogenetic T stages	
(p)oT1	46 (79%)
(p)oT2	12 (21%)
Infiltrated extracervical tissues:	
Lateral parametria	5 (9%)
Uterine corpus	6 (10%)
Vaginal fornix/wall	1 (2%)

^1^ BMI, body mass index; ^2^ LVSI, lympho-vascular space involvement; ^3^ LNs, lymph nodes; ^4^ Maximal tumor size of the TMMR specimen if no conization or diameters of the conization and TMMR specimens.

**Table 3 diagnostics-11-01749-t003:** Imaging data.

Imaging Parameters	Ultrasound (*n* = 58)	Magnetic Resonance (*n* = 50)
Route		
Transvaginal, *n* (%)	24 (41%)	does not apply
Transrectal, *n* (%)	34 (59%)	
Tumor (primary or residual post conization) detactable		
Yes, *n* (%)	49 (84%)	41 (82%)
No, *n* (%)	9 (16%)	9 (18%)
Maximal tumor size, mm, median (IQR)	25 (26.5)	24 (28.5)
Lateral parametria involvement	0	10 (20%)
Uterine corpus infiltration	5 (9%)	7 (14%)
Mesometrial LNs detectable	1 (2%)	0
Vaginal wall/fornix infiltration	1 (2%)	13 (26%)
Anterior to the cervix involvement	0	0
Posterior to the cervix involvement	0	2 (4%)
FIGO 2018 stage		
IB1	19 (33%)	21 (42%)
IB2	19 (33%)	5 (10%)
IB3	19 (33%)	4 (8%)
IIA1	0	3 (6%)
IIA2	1 (1%)	6 (12%)
IIB	0	9 (18%)
>IIB	0	2 (4%) (IVA)
T stage, from TNM system, 8th ed., 2016		
1b1 (< 4 cm)	38 (66%)	26 (52%)
1b2 (> 4 cm)	19 (33%)	4 (8%)
2a1	0	3 (6%)
2a2	1 (1%)	6 (12%)
2b	0	9 (18%)
>2b	0	2 (4%) (4a)
Ontogentic T stage (preoperatively)		
oT1	53 (91%)	30 (60%)
oT2	5 (9%)	18 (36%)
oT3	0	2 (4%)

**Table 4 diagnostics-11-01749-t004:** Accuracy of ultrasonography for tumor detection and localizations of the disease (*n* = 58).

Ultrasonography Assessment	Sensitivity, %	Specificity, %	PPV ^1^, %	NPV ^2^, %	Accuracy, %
Tumor detectable ^3^	94.12 (87.66–100.00)	85.71 (59.79–100.00)	97.96 (94.00–100.00)	66.67 (35.87–97.46)	93.10 (86.58–99.62)
Parametrial involvement	0 (0.00–0.00)	100 (100.00–100.00)	NC ^4^	89.66 (81.82–97.49)	89.66 (81.82–97.49)
Uterine corpus involvement	66.67 (28.95–100.00)	98.08 (94.34–100.00)	80.00 (44.94–100.00)	96.23 (91.10–100.00)	94.83 (89.13–100.00)
Vaginal fornix/wall involvement	0 (0.00–0.00)	98.21 (94.75–100.00)	0 (0.00–0.00)	96.49 (91.71–100.00)	94.83 (89.13–100.00)

^1^ PPV, positive predictive value; ^2^ NPV, negative predictive value; ^3^ The primary tumor after biopsy or the residual tumor after conization; ^4^ NC, not calculable; lower and upper confidence interval presented in brackets.

**Table 5 diagnostics-11-01749-t005:** Accuracy of ultrasonography and MRI for tumor detection and localizations of the disease (*n* = 50).

Imaging Assessment		Sensitivity, %	Specificity, %	PPV ^1^, %	NPV ^2^, %	Accuracy, %	*p*-Value ^3^
Tumor detectable ^4^	Ultrasonography	93.18 (85.73–100.00)	83.33 (53.51–100.00)	97.62 (93.00–100.00)	62.50 (28.95–96.05)	92.00 (84.48–99.52)	0.56
	MRI	90.91 (82.42–99.40)	83.33 (53.51–100.00)	97.56 (92.84–100.00)	55.56 (23.09–88.02)	90.00 (81.68–98.32)	
Parametrial involvement	Ultrasonography	0 (0.00–0.00)	100 (100.00–100.00)	NC ^5^	92.00 (84.48–99.52)	92.00 (84.48–99.52)	0.32
	MRI	25.00 (0.00–64.44)	80.43 (68.97–91.90)	10.00 (0.00–28.59)	92.50 (84.34–100.00)	76.00 (64.16–87.84)	
Uterine corpus involvement	Ultrasonography	66.67 (28.95–100.00)	97.73 (93.32–100.00)	80.00 (44.94–100.00)	95.56 (89.53–100.00)	94.00 (87.42–100.00)	0.32
	MRI	50.00 (9.99–90.00)	90.91 (82.42–99.40)	42.86 (6.20–79.52)	93.02 (85.41–100.00)	86.00 (76.3895.62)	
Vaginal fornix/wallinvolvement	Ultrasonography	0 (0.00–0.00)	97.96 (94.00–100.00)	0 (0.00–0.00)	97.96 (94.00–100.00)	96.00 (90.57–100.00)	0.32
	MRI	100 (100.00–100.00)	75.51 (63.47–87.55)	7.69 (0.00–22.18)	100 (100.00–100.00)	76.00 (64.16–87.84)	

^1^ PPV, positive predictive value; ^2^ NPV, negative predictive value; ^3^ determined using McNemar’s test (*n* = 50); ^4^ The primary tumor after biopsy or the residual tumor after conization; ^5^ NC, not calculable; lower and upper confidence interval presented in brackets.

**Table 6 diagnostics-11-01749-t006:** Discordance between imaging (ultrasound and MRI) and histology for primary and residual tumor (after conization) measurements (additional data in Appendix A).

Tumor Size	Ultrasonography Mean Error	MRI Mean Error	Ultrasonography Better ^1^ *n* (%)	MRI Better ^2^ *n* (%)	Ultrasonography and MRI Equal ^3^ *n* (%)
By maximal size, mm	5.70	8.84	20 (40%)	22 (44%)	8 (16%)
By mean size, mm	4.6	5.9	21 (42%)	20 (40%)	9 (18%)
By volume ^4^, mm^3^	11775	15518	20 (40%)	22 (44%)	8 (16%)
By substitutive diameter ^5^	5.9	7.4	20 (40%)	22 (44%)	8 (16%)

^1^ Number of patients who had a smaller measurement difference between ultrasonography and histology than between MRI and histology; ^2^ Number of patients who had a smaller measurement difference between MRI and histology than between ultrasonography and histology; ^3^ Number of patients with the same measurement difference between ultrasound and histology and MRI and histology; ^4^ Volume calculated from the three available measurements (i.e., length, depth, and width), not from imaging device software); ^5^ Definition provided in the Methods section.

**Table 7 diagnostics-11-01749-t007:** Accuracy of ultrasonography and MRI for cervical cancer stage prediction ^1^.

Stage Classification	Ultrasound (*N* = 58)	MRI (*N* = 50)	*p*-Value ^5^
FIGO 2018 ^2^	68.97% (57.06–80.87%)	42.00% (28.32–55.68%)	0.0016
T ^3^	79.31% (68.89–89.74%)	52.00% (38.15–65.85%)	0.0005
oT ^4^	87.93% (79.55–96.31%)	70.00% (57.30–82.70%)	0.0045

^1^ Detailed data are available in Appendix A; ^2^ staging classification by the International Federation of Gynecology and Obstetrics 2018; ^3^ staging classification by tumor (T) stage of the TNM system; ^4^ staging classification by ontogenetic staging by Hockel et al. (see Appendix A); ^5^ determined using McNemar’s test (*n* = 50); lower and upper confidence interval presented in brackets.

## Data Availability

The study protocol can be accessed upon demand of interested researchers if justified. This study (about the accuracy of imaging) was not registered in any repository. The TMMR-RS is registered on ClinicalTrials.gov (NCT01819077).

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
