# Peer review of "Accuracy of Ultrasonography and Magnetic Resonance Imaging for Preoperative Staging of Cervical Cancer—Analysis of Patients from the Prospective Study on Total Mesometrial Resection"

_diagnostics, 2021, doi:10.3390/diagnostics11101749_

Round 1
Reviewer 1 Report
This was a retrospective study intended to evaluate the diagnostic performance of MRI and ultrasound in the preoperative evaluation of cervical cancer patients. This is an important and clinically relevant study. However, some points, especially regarding the statistical analysis, need to be addressed.
General remark:
Throughout the text, please ensure that “.” and “,” are used appropriately (e.g. 1.5% instead of 1,5%), example: line 658.
The Materials and Methods, Results, and Discussion section should be reviewed meticulously for grammatical errors.
Abstract:
- The conclusion in the abstract is not supported by the research (see below)
- To keep the abstract short, I would not discuss regional staging (Lymph nodes) here.
Introduction:
- Well written, clear and informative introduction.
- Page 2 line 78: the authors should write “the primary objectives of the PRESENT study” or “the primary objectives of the study presented here” to clarify that they refer to their own study presented.
Materials and Methods:
- In this section, there are numerous minor grammar errors. I will give some examples throughout.
- Please state right from the beginning that the 2018-FIGO classification was used
- Page 3 line 88: “and thus” instead of “and, thus”
- The trial design should be described more clearly at the beginning of the methods section. Furthermore, it should be stated explicitly that this investigation was not part of the TMMR-RS. For example: “This is a single-site subgroup analysis of the prospective, observational TMMR register study including all consecutive study patients from Gdynia Oncology Center”.
- In the trial protocol published at clinicaltrials.gov, no preoperative imaging requirements are specified. This should be clear in the text, for example: “Though not required as part of the TMMR-RS, most patients received preoperative ultrasound as part of the routine work up and the results were collected at the investigation site”.
- Though the authors mention this later on, it should be made clear already at this point that all patients were recruited from a single site (Gdynia Oncology Center) as the register study has a multicentric layout.
- Page 3 line 117: “LN regions” instead of “LNs regions”.
- Line 122: This paragraph is a bit confusing. Maybe it would be simpler to state that “As part of the TMMR-RS, some patients underwent additional abdominal and pelvic MR-imaging, although not explicitly required in the study protocol”.
- Line 123: “ and imaging with ultrasound” instead of “, imaging with ultrasound”.
- Line 177: “the cancer” or “the tumor” instead of “the cancer tumor”
- Line 229: “The areas anterior and posterior” instead of “Area anterior and posterior”
- Line 263: “the aortic bifurcation” instead of “the aorta bifurcations”
- Line 275: “was scanned” instead of “were scanned”
Statistical analysis:
- In my opinion, the main problem of this study is the statistical analysis. No comparative tests regarding the imaging modalities were carried out, which is essential in order to calculate statistical significance of the findings. Presenting just percentages from two groups without comparing them is not adequate, and the conclusions the authors draw from their findings (that ultrasound is superior to MRI) cannot be supported on a scientifically sound basis.
- The authors write that they use a confusion matrix to compare the two imaging methods, however, it seems that the confusion matrix was only used to compare each imaging method to the reference test (histology) and not to compare them to each other. Such confusion matrixes can be used to perform e.g. McNemar tests to compare sensitivity and specificity between the imaging methods and to calculate the statistical significance between these modalities. As the authors used R for statistical calculations, I suggest using the DTComPair and exact2x2 packages for comparison of sensitivity, specificity, PPV, NPV and accuracy. This also provides confidence intervals and p-values.
- Line 397: I assume that the authors meant to write “multivariable analysis” instead of “variability testing”.
- The authors mention ontogenetic staging several times. I think many readers are lost with this. It would be helpful to add a supplementary table outlining the ontogenetic tumor stages.
Results:
- Although the authors state that they wrote the manuscript in accordance with the STARD criteria, not all information as outlined int the STARD protocol is presented. A standardized STARD flow chart in the appendix would be useful, but is not mandatory.
- Table 1: giving interquartile ranges (IQR) instead of absolute ranges is more informative for continuous data.
- Table 4: in the caption “NC” instead of “NM”.
- Line 472: the false positive rate is 0.2, not 90%. FPR = FP/(FP+TN)
- Line 511: confirmed instead of conformed
- Line 522: recalculate the false positive rate as given above
- Line 533: “multivariable analysis” instead of variability analysis, I assume. Personally, I think that there should be enough data to perform univariable analysis for several parameters, but it is absolutely acceptable to forego this. In that case, I would remove the entire paragraph from the text.
Discussion:
- Line 605: this statement cannot be made without statistical comparison of the groups.
- Line 620: same issue as before
- The following paper might be useful for discussion of parametrial involvement: Sodeikat et al., Cancers 2021, DOI: 3390/cancers13122961
Conclusion:
- As stated before, the conclusion that ultrasound is better than MRI (e.g. in stage prediction) cannot be drawn without appropriate statistical tests.
Author Response
diagnostics-1341194
Title: Accuracy of ultrasonography and magnetic resonance imaging for preoperative staging of cervical cancer – analysis of patients from the prospective study on total mesometrial resection.
Dear Reviewer,
Thank you very much for your opinion and valuable comments on our article. We very much appreciate your experience in the field, and that you had time and took the effort to review the manuscript.
At most, thank you very much for your detailed, professional review. Especially, thank you for the most important comments regarding the statistical analysis and the presented results. We apologize very much for not including these crucial data in the submitted manuscript. The authors made appropriate calculations, and we incorporated them into the manuscript and provided adequate results and conclusions.
Our detailed responses are below your comments:
Comments and Suggestions for Authors
This was a retrospective study intended to evaluate the diagnostic performance of MRI and ultrasound in the preoperative evaluation of cervical cancer patients. This is an important and clinically relevant study. However, some points, especially regarding the statistical analysis, need to be addressed.
Answer: Thank you … However, our study was not a retrospective one. We mentioned it in the manuscript: „This is a sub-analysis from the prospective, observational clinical trial”, and „Data on ultrasonography and MRI were prospectively collected before surgery, saved, and not changed thereafter.” – Section 2.1. It is true that the TMMR-RS primary objectives are different – not imaging issues, nevertheless, all data and methodology for this sub-analysis (focused on imaging) was performed prospectively.
General remark:
Throughout the text, please ensure that “.” and “,” are used appropriately (e.g. 1.5% instead of 1,5%), example: line 658.
Answer: Thank you for this remark. You are right of course. I tried to keep the form „1.5” and not the 1,5), throughout the text, but must have missed a few. This issue came because we operate the form „1,5” in my country, while it should be „1.5” of course when writing in English. We apologize for this issue. I checked the text and modified the mistakes. Moreover, we sent the text for professional English language editing.
In line 658 or around, there were no digits, just the text. So I cannot see the example. However, I found 3 of these kinds of mistakes in lines 693-694 of the manuscript (and corrected these of course).
The Materials and Methods, Results, and Discussion section should be reviewed meticulously for grammatical errors.
Answer: We have sent the text for professional English language editing, so the revised manuscript is now professionally checked for English language style, grammar, and spelling. These changes were linguistic only and thus were not left in the “track changes” mode. We only left changes that we made according to the Review’s comments and suggestions, to let her/him know about the corrections.
Abstract:
- The conclusion in the abstract is not supported by the research (see below)
Answer: We corrected it accordingly. See below.
- To keep the abstract short, I would not discuss regional staging (Lymph nodes) here
Answer: Thank you for this comment. I agree. The text about lymph nodes was removed from the abstract.
Introduction:
- Well written, clear, and informative introduction.
Answer: Thank you.
- Page 2 line 78: the authors should write “the primary objectives of the PRESENT study” or “the primary objectives of the study presented here” to clarify that they refer to their own study presented.
Answer: We added the suggested words. I agree that it is necessary to clarify, that our study was separate / was a sub-analysis for another, primary study, namely the TMMR-RS.
Materials and Methods:
- In this section, there are numerous minor grammar errors. I will give some examples throughout.
Answer: We apologize very much for these errors. It should not happen. All of the authors read the text, everyone corrected some issues, but still, errors appeared. We have sent the text for professional English language editing, so the revised manuscript is now professionally checked (the proof can be sent).
- Please state right from the beginning that the 2018-FIGO classification was used
Answer: We stated that the 2018-FIGO classification was used in sections: 2.3 and 2.3.4.2 – these sections refer directly to our study objectives and design (about the imaging). We did not state that the 2018-FIGO classification was used in earlier sections, because we described patients included in the TMMR-RS (section 2.2). The TMMR-RS was launched in March 2013 (ClinicalTrials.gov Identifier: NCT01819077). At that time there was the 2009-FIGO classification. Thus, we cannot state that the 2018 classification was used right at the beginning. Moreover, we stated that patients with IB-IIA cervical cancer were included (there were no restrictions to the tumor size); no stratification within the IB stage (for the TMMR-RS study). Exclusion criteria: all IA, and >IIA. We acknowledge, that it is somewhat complicated, however, the FIGO classification was changed during the TMMR-RS. Moreover, we wrote the manuscript in the 2021 year, so we considered it appropriate to refer and to and use the 2018 classification.
- Page 3 line 88: “and thus” instead of “and, thus”
Answer: corrected.
- The trial design should be described more clearly at the beginning of the methods section. Furthermore, it should be stated explicitly that this investigation was not part of the TMMR-RS. For example: “This is a single-site subgroup analysis of the prospective, observational TMMR register study including all consecutive study patients from Gdynia Oncology Center”.
Answer: Thank you. We have clarified this issue according to the Reviewer’s suggestion. It is written at the beginning of the methods section, and in the abstract.
- In the trial protocol published at clinicaltrials.gov, no preoperative imaging requirements are specified. This should be clear in the text, for example: “Though not required as part of the TMMR-RS, most patients received preoperative ultrasound as part of the routine work up and the results were collected at the investigation site”.
Answer: Thank you. We have corrected it as suggested.
- Though the authors mention this later on, it should be made clear already at this point that all patients were recruited from a single site (Gdynia Oncology Center) as the register study has a multicentric layout.
Answer: It was clarified in the first sentence of the methods section (2.1), and the abstract.
- Page 3 line 117: “LN regions” instead of “LNs regions”.
Answer: Thank you. We have corrected it.
- Line 122: This paragraph is a bit confusing. Maybe it would be simpler to state that “As part of the TMMR-RS, some patients underwent additional abdominal and pelvic MR-imaging, although not explicitly required in the study protocol”.
Answer: Thank you. We have corrected it as suggested.
- Line 123: “ and imaging with ultrasound” instead of “, imaging with ultrasound”.
Answer: Thank you. We have corrected it.
- Line 177: “the cancer” or “the tumor” instead of “the cancer tumor”
Answer: Thank you. We have corrected it.
- Line 229: “The areas anterior and posterior” instead of “Area anterior and posterior”
Answer: Thank you. We have corrected it. Additionally, the English language editing office corrected some of these statements.
- Line 263: “the aortic bifurcation” instead of “the aorta bifurcations”
Answer: Thank you. We have corrected it.
- Line 275: “was scanned” instead of “were scanned”
Answer: Thank you. We have corrected it.
Statistical analysis:
- In my opinion, the main problem of this study is the statistical analysis. No comparative tests regarding the imaging modalities were carried out, which is essential in order to calculate statistical significance of the findings. Presenting just percentages from two groups without comparing them is not adequate, and the conclusions the authors draw from their findings (that ultrasound is superior to MRI) cannot be supported on a scientifically sound basis.
- The authors write that they use a confusion matrix to compare the two imaging methods, however, it seems that the confusion matrix was only used to compare each imaging method to the reference test (histology) and not to compare them to each other. Such confusion matrixes can be used to perform e.g. McNemar tests to compare sensitivity and specificity between the imaging methods and to calculate the statistical significance between these modalities. As the authors used R for statistical calculations, I suggest using the DTComPair and exact2x2 packages for comparison of sensitivity, specificity, PPV, NPV and accuracy. This also provides confidence intervals and p-values.
Answer: Thank you for your very valuable feedback. Indeed, comparing the two methods in terms of statistics will significantly improve the quality of the results presented. As recommended by the Reviewer, we used the McNemar’s test to compare both methods. McNemar’s test is a paired nonparametric or distribution-free statistical hypothesis test. The McNemar’s test is checking if the disagreements between two cases match. Technically, this is referred to as the homogeneity of the contingency table (specifically the marginal homogeneity). Therefore, the McNemar’s test is a type of homogeneity test for contingency tables.
Appropriate additions have been added to the “2.5. Statistical analysis” section and to tables 5 and 7.
- Line 397: I assume that the authors meant to write “multivariable analysis” instead of “variability testing”.
Answer: The „variability” nomenclature is taken from the STARD criteria, thus it was intentionally used - see number 17 in the Table 1 in the STARD 2015 publication (BMJ 2015;351:h5527 doi: 10.1136/bmj.h5527). We did not want to perform a multivariable analysis.
- The authors mention ontogenetic staging several times. I think many readers are lost with this. It would be helpful to add a supplementary table outlining the ontogenetic tumor stages.
Answer: The ontogenetic staging is described elsewhere, and the citation is provided. Nevertheless, I agree, that the ontogenetic staging should be provided for the readers, in a form of a supplementary table. I added the ontogenetic staging in a supplementary table.
Results:
- Although the authors state that they wrote the manuscript in accordance with the STARD criteria, not all information as outlined int the STARD protocol is presented. A standardized STARD flow chart in the appendix would be useful, but is not mandatory.
Answer: At the final version of the manuscript one may not clearly see the application of STARD criteria. These are not presented in points. However, I must assure, that the manuscript was started and built on the frame of all STARD criteria, points, as outlined in the cited reference. The STARD structure was later „covered” by the text, which was later even more edited to make the text readable (flow of the text). During the process of editing and reviewing the text, none of the STARD information was delated, at least intentionally. Thus, all required STARD information should be there, and we think that an additional table outlining STARD structure would be a repetition of the already provided information in the main manuscript.
- Table 1: giving interquartile ranges (IQR) instead of absolute ranges is more informative for continuous data.
Answer: We guess that the Reviewer meant Table 2. We added IQR ranges wherever possible and rational.
- Table 4: in the caption “NC” instead of “NM”.
Answer: Thank you. We apologize for the error. We have corrected it.
- Line 472: the false positive rate is 0.2, not 90%. FPR = FP/(FP+TN)
Answer: FPR value was recalculated
- Line 511: confirmed instead of conformed
Answer: Thank you. We apologize for the error. We have corrected it.
- Line 522: recalculate the false positive rate as given above
Answer: FPR value was recalculated
- Line 533: “multivariable analysis” instead of variability analysis, I assume. Personally, I think that there should be enough data to perform univariable analysis for several parameters, but it is absolutely acceptable to forego this. In that case, I would remove the entire paragraph from the text.
Answer: The term „variability analysis” was used intentionally, because the STARD criteria required this – see number 17 in the Table 1 in the STARD 2015 publication (BMJ 2015;351:h5527 doi: 10.1136/bmj.h5527).
Discussion:
- Line 605: this statement cannot be made without statistical comparison of the groups.
Answer: Line 605 is in the Results section (not Discussion), in the manuscript downloaded from the Diagnostics, after the review. It presents just descriptive results.
- Line 620: same issue as before
Answer: Line 620 (in the manuscript downloaded after the review) represents just a part of the summarized results from our study. Nevertheless, the whole paragraph (including the 620 line) was reorganized, accordingly to more precise results.
- The following paper might be useful for discussion of parametrial involvement: Sodeikat et al., Cancers 2021, DOI: 3390/cancers13122961
Answer: Thank you for this reference. I have used it in the discussion section.
Conclusion:
- As stated before, the conclusion that ultrasound is better than MRI (e.g. in stage prediction) cannot be drawn without appropriate statistical tests.
Answer: Thank you very much. Of course, we agree. We added suggested calculations in the Results section, and accordingly provided conclusions. The paragraph is now changed, and conclusions are drawn from appropriate statistical tests.
Yours sincerely,
in the name of all authors,
Maciej Stukan
Reviewer 2 Report
The manuscript is well written, however
the population of the study is too small and
the subject not so original.
Author Response
diagnostics-1341194
Title: Accuracy of ultrasonography and magnetic resonance imaging for preoperative staging of cervical cancer – analysis of patients from the prospective study on total mesometrial resection.
Dear Reviewer,
Thank you very much for your opinion and valuable comments on our article. We very much appreciate your experience in the field, and that you had time and took the effort to review the manuscript.
Our response is below your comments:
Comments and Suggestions for Authors:
The manuscript is well written,
Answer: Thank you very much.
The population of the study is too small
Answer: There is an interim analysis of the TMMR-RS being performed at the moment. It is done by primary investigators. This was the reason to perform this additional study, a sub-analysis about imaging, independent of the main trial results and outcomes. We encouraged other participating institutions to join the study, however it appeared, that the ultrasonography was not performed with that systematic approach as we assumed for the study. The preoperative imaging method is not described in the TMMR-RS inclusion/exclusion criteria. The investigators may use another preoperative workup before the TMMR/tLND.
The subject not so original
Answer: I agree, that some publications about the usage of ultrasonography for cervical cancer staging were published a few years ago. Nevertheless, still, the MRI is considered a gold standard, and most clinicians are rather sceptic about the value of expert ultrasonography, despite it was mentioned in the international guidelines, that expert ultrasonography can be used for preoperative workup. We considered it worth working on this subject and present our results, with additional figures and videos that add an educational value to the article (all could be freely downloaded by readers if published in Diagnostics (MDPI)).
Yours sincerely,
in the name of all authors,
Maciej Stukan
Round 2
Reviewer 1 Report
The authors did a great job in improving the paper. There are still a few minor typos (e.g. page 21 line 631: predict instead of prediction), but overall it seems fine. The results are now backed by a well performed statistical analysis and the conclusions the authors draw are correct.